# Physicochemical Water Quality Influence on the Parasite Biodiversity in Juvenile Tilapia (*Oreochromis* spp.) Farmed at *Valle Del Mezquital* in the Central-Eastern Socioeconomic Region of Mexico

**DOI:** 10.3390/pathogens11101076

**Published:** 2022-09-21

**Authors:** Víctor-Johan Acosta-Pérez, Vicente Vega-Sánchez, Tomás-Eduardo Fernández-Martínez, Andrea-Paloma Zepeda-Velázquez, Nydia-Edith Reyes-Rodríguez, Jesús-Benjamín Ponce-Noguez, Armando Peláez-Acero, Jorge-Luis de-la-Rosa-Arana, Fabián-Ricardo Gómez-De-Anda

**Affiliations:** 1Área Académica de Medicina Veterinaria y Zootecnia, Instituto de Ciencias Agropecuarias, Tulancingo de Bravo 43600, Hidalgo, Mexico; 2Área Académica de Medicina, Instituto de Ciencias de la Salud, Pachuca 42090, Hidalgo, Mexico; 3Medicina Veterinaria, Universidad de la Costa, Pinotepa Nacional 71600, Oaxaca, Mexico; 4Microbiología en Salud Humana, Facultad de Estudios Superiores Cuautitlán, Universidad Nacional Autónoma de Mexico, Cuautitlan Izcalli, 54743, Estado de Mexico, Mexico

**Keywords:** *Oreochromis* spp., parasitosis, environment, correlation, aquaculture

## Abstract

Aquaculture parasite biodiversity dependents on multiple environmental characteristics, including water quality. The analysis of this relationship aims to support improvements in the production management of tilapia. For this purpose, a total of 100 juvenile fishes (*Oreochromis* spp.) and 30 water samples were collected at *Valle del Mezquital* in the Central-Eastern socioeconomical region of Mexico. A study of parasite biodiversity was carried out and water quality parameters were determined. Biodiversity in the habitat was measured using the Simpson diversity index, which considers the number of species present and the abundance of each one; we also calculate the Berger-Parker index to estimate the proportional importance of the most abundant species. In general, it was found that 86% of the examined specimens were parasitized. Parasite biodiversity was 11 genera (Simpson index = 0.55). *Trichodina* spp. (*Ciliophora*) was the dominant genus (Berger-Parker index = 0.51). The protozoa *Apiosoma* spp. was associated with the water hardness (Berger-Parker index = 0.57). Furthermore, the presence of monogeneans showed a positive correlation with the levels of nitrites and ammonium in the water (Berger-Parker index = 0.06–0.55). This characterization may represent a useful tool in the comprehensive management of parasites that affect the farmed tilapia. However, new data is necessary to expand the knowledge about the environment-host-parasite relationship.

## 1. Introduction

Tilapia belongs to a group of African-origin fishes with socio-economic importance that are classified in the *Cichlidae* family. The genus *Oreochromis* groups 32 species; among them, *O. niloticus* is the most frequently farmed in more than 90 countries around the world [1,2]. Tilapia production represents a food industry with excellent prospects for obtaining high-quality animal protein. In fact, 7.3 million tons of tilapia meat is estimated to be produced worldwide by 2030 [3], because tilapia can easily adapt to environmental and food changes [4,5,6]. In addition, tilapia is resistant to diseases of microbiological and parasitological origin [4]. Actually, parasites can act as part of microbiota in the biology of the individual, in the population structure and in the functioning of ecosystems in aquaculture production systems [7]. However, it has been documented that large-scale environmental variations, such as climate change [8] and anthropocentric actions [9], induce modifications in the distribution and biodiversity of tilapia parasites, which could ultimately affect the human and veterinary health [10,11]. Regarding parasitic infections, protozoans (*Ichthyophthirius* spp., *Trichodina* spp., *Chilodonella* spp., *Ichtyobodo* spp.), monogeneans (*Cichlidogyrus* spp., *Gyrodactylus* spp., *Dactylogyrus* spp.), and nematodes (*Contracaecum* spp.) have been described in tilapias [12,13,14,15]. It is worth noting that parasites represent a latent risk for the development of diseases that can cause economic losses in aquaculture production [16]. This work was carried out at *Valle del Mezquital*, a region located in Central-Eastern Mexico, which represents an area of high production of tilapia. However, *Valle del Mezquital* receives millions of cubic meters of untreated wastewater from Mexico City and its metropolitan area, a health risk for the aquaculture production systems found in the region [17]. Thus, the objective of this work was to analyze the influence of physicochemical water quality on the parasitic biodiversity of juvenile farmed tilapias at *Valle del Mezquital*.

## 2. Results

### 2.1. Frequency

The analyzed fishes presented a body conformation index of 1.52 ± 0.42 to 2.82 ± 0.87. In general, it was observed that 86.00% (95% CI 84.26–87.74) of the study animals presented some type of parasite. In the work, here report that 6862 parasites were observed, corresponding to 11 genera listed in Table 1. The footnote of Table 1 shows the decoding of acronyms that corresponds to the farm identification where the tilapia collection was performed. The monogenean *Dactylogyrus* spp. was the one found most frequently (52.00% 95% CI 49.65–54.35). The protozoan *Trichodina* spp. was the second most frequent parasite in the study here reported (42.00% 95% CI 39.76–44.24). In addition, six other genera of protozoa were found, including *Apiosoma* spp. (12.00% 95% CI 9.90–14.10), *Chilodonella* spp. (3.00% 95% CI 2.54–3.46), *Costia* spp. (2.00% 95% CI 1.72–2.28), *Scyphydia* spp. (1.00% 95% CI 0.79–1.21), and *Tetrahymena* spp. (6.00% 95% CI 5.09–6.91). Likewise, dinoflagellate protozoa of genus *Oodinium* spp. (2.00% 95% CI 1.72–2.28) were identified. Figure 1 shows representative specimens of protozoa identified in this study. In addition to *Dactylogyrus*, other monogeneans found were *Dawestrema* spp. (11.00% IC95% 9.60–12.40), *Cichlidogyrus* spp. (4.00% IC95% 3.15–4.85) and *Gyrodactylus* spp. (14.00% IC95% 12.54–15.46). Figure 2 shows these monogeneans.

### 2.2. Distribution and Biodiversity

Sampling points G9Te and G4Ch showed the highest parasitic presence with 4325 and 1954 individuals (total sum of parasites regardless of genus). Table 2 shows that 91.5% of recovered parasites were from these two farms. Data obtained indicate that the parasitic biodiversity in *Oreochromis* spp. was between 2 and 10 genera per sampling site. The biodiversity in the different sampling locations showed a global Simpson index of 0.55. However, values higher than the average were observed among locations; point G1Ix presented a Simpson index of 0.60, followed by point G2Ix with an index of 0.59, while points G5Ix and G7Te represented the points with the smallest parasitic diversity with Simpson indices of 0.05 and 0.02, respectively. The protozoan *Trichodina* spp. was the globally dominant genus (Berger-Parker index = 0.51). At point G9Te, the protozoan *Apisoma* spp. was predominant over nine parasitic genera present in the sample (dominance index = 0.66). Finally, the monogenean *Dactylogyrus* spp. was dominant in five sampling points.

### 2.3. Ecological Indices of Abundance and Intensity

To characterize the parasite-host relationship, the ecological indices of abundance and intensity were calculated. The ecological index of abundance (No. of parasites/No. of fish sampled) showed that the protozoa *Trichodina* spp. (35.29) and *Apisoma* spp. (29.00) were the parasites with the highest presence in the tilapia population sampled in this study. *Dactylogyrus* spp. was the monogenean (2.21) with the highest abundance, while the eight remaining genera presented indices between 0.03 and 1.19 (Figure 3a). In this study, the genera of protozoa *Apiosoma* spp. (241.66) and *Trichodina* spp. (84.02) showed the highest intensity of infection (No. of parasites/No. of infected fish). On the other hand, among the monogeneans, *Dactylogyrus* spp. presented the highest intensity of infection with an index of 4.25, while the remaining eight genera presented values between 1.50 and 39.66 (Figure 3b).

### 2.4. Water Quality

The water quality was evaluated at the 10 sampling points, and the mean (±standard deviation) of the physicochemical parameters is found in Table 3. The pH values found were in the range of 6.46 ± 0.05 to 7.46 ± 0.05 for the various locations. The high concentration of CO_2_ demonstrated poor water quality in farm G3Ix (650 ± 17.32 mg mL^−1^), followed by farms G5Ix and G8Te with a CO_2_ concentration of 590 ± 121.24 and 290 ± 17.32 mg mL^−1^, respectively. The measurement of nitrogenous compounds showed that the levels of nitrites (NO_2-_-N) were between 0 and 1.00 ± 0 mg L^−1^, with higher values in the farms corresponding to the points from G5Ix to G10Pr (*p* = 0.000). The concentration of nitrates (NO_3-_-N) was between 0 and 33.33 ± 11.54 mg L^−1^. In farm G5Ix, the highest concentration of nitrates was found (*p* = 0.000). The ammonium concentration (NH_3_-N) found was between 0.50 ± 0 to 2.50 ± 0 mg L^−1^; in farm G1Ix the highest concentration of ammonium was found (*p* = 0.000). Finally, the concentration of calcium carbonates showed values of up to 511 ± 7.54 mg L^−1^, while the alkalinity values of the water were up to 507 ± 27.49.

### 2.5. Influence of Water Quality on Parasite Biodiversity

The correlation index between parasite prevalence and physicochemical parameters of water quality was analyzed using a Pearson correlation matrix. The correlation index of the protozoa *Chilodonella* spp. and *Apisoma* spp. was positive, with the concentrations of nitrites, ammonium, hardness, and alkalinity in water quality (between 0.23 and 0.57; *p* < 0.05), where the positive correlation of both protozoa with the concentration of calcium carbonates (between 0.53 and 0.57; *p* < 0.05) stands out. Furthermore, in this study the genus *Tetrahymena* spp. presented a positive correlation with CO_2_ concentrations (0.55). This analysis allowed us to observe a positive correlation index between the prevalence of *Trichodina* spp. and the rest of the protozoa identified in this work (between 0.27 and 0.92; *p* < 0.05), highlighting the positive correlation that it maintains with the prevalence of *Costia* spp. (0.92) and *Oodinium* spp. (0.92), whereas the highest correlation index between populations of protozoan parasites was between *Costia* spp. and *Oodinium* spp. with a value of 1 (Figure 4).

Figure 5 shows that the correlation analysis for the prevalence of monogeneans showed a positive association between the genera *Dawestrema* spp., *Cichlidogyrus* spp. and *Gyrodactylus* spp., with the concentration of CaCO_3_ (0.83, 0.51 and 0.58 respectively; *p* < 0.05). A positive correlation was also observed between these genera and nitrites (NO_2_- -N mg L^−1^) and ammonium (NH_3_- N mg L^−1^), with indices between 0.06 and 0.55; *p* < 0.05. *Dactylogyrus* spp. showed no positive associations with any parameter of water quality. However, it showed negative correlations with nitrogenous compounds with indices from −0.06 to −0.43. Finally, the prevalence of the monogenean *Gyrodactylus* spp. presented the highest positive correlation rates with other monogenean infections (0.40 and 0.74; *p* < 0.05), highlighting its association with *Dawestrema* spp. and *Cichlidogyrus* spp.

## 3. Discussion

### 3.1. Frequency

The 86% of parasite prevalence in our study was higher than that reported by Pantoja et al. in 2012 [15], who reported a parasitic frequency of 64.2% in tilapia farmed in Brazil. The prevalence determined for the monogenean *Dactylogyrus* spp. agrees with that reported in other countries with aquaculture production, such as India, where monogeneans have been reported with a prevalence of 26% in tilapias farmed [18]. Similarly, infection by *Trichodina* spp. in tilapias farmed is a constant health risk since infection frequencies have been recurrently reported, ranging from 37.5 to 96.3% [7,19,20]. Furthermore, *Apiosoma* spp., *Chilodonella* spp., *Costia* spp., *Scyphydia* spp. and *Tetrahymena* spp. are well known as parasites that attack tilapia populations [21]. It should be noted that the frequency of infection reported here is higher than that observed in studies in free-living fish [14]. This suggests a focus on the attention of the parasitosis that occur in fishes farmed.

One of the limitations of our study is that we did not look for the presence of viruses or bacteria because it was not the objective of the work, although we are aware that microbiological contamination is responsible for more than 90% of intoxications and the transmission of diseases by water. It is important to consider that the presence of pathogenic microorganisms in drinking water is the source of infection for numerous diseases for both public and veterinary health. The identification of microorganisms in tilapia ponds, in addition to being informative, could guide production practices.

The prevalence in our study determined important parasites (*Dactylogyrus* spp., *Apiosoma* spp., *Chilodonella* spp., *Trichodina* spp. *Costia* spp., *Scyphydia* spp. and *Tetrahymena* spp.) with higher infections in fish farms than in free-living fish.

### 3.2. Distribution and Biodiversity

The sampling localities with the highest frequency of infection corresponded to the Tezontepec de Aldama and Chilcuautla municipalities. According to the distribution, the genus *Trichodina* spp. was identified in the 10 sampling points, covering the four sampled municipalities. In the same way, the monogenean *Dactylogyrus* spp. was found in the four municipalities under study, but only in nine of the 10 sampled localities, in contrast to the protozoa *Scyphydia* spp. and the monogenean *Cichlidogyrus* spp. that were identified in a single point each, Chilcuautla and Tezontepec de Aldama, respectively.

The parasite diversity in fishes were from 11 different genera, similar to the 11 parasites previously reported in *O. niloticus* from Kenya; that study reported nine parasites at the genus level and two at the species level [22]. The parasitic diversity calculated by the Simpson index showed a value lower than the 0.88 reported in the Turkana fishing environments in Kenya for *Tilapia zillii* [23]. In addition, the low biodiversity values that some sampling points presented may be associated with the presence of a dominant parasite. In the case of points G5Ix and G7Te, it was the protozoan *Trichodina* spp. with a Berger-Parker index of *d* = 0.97 and *d* = 0.98, respectively [24]. This protozoan was the dominant species in four sampling locations: G4Ch, G5Ix, G6Te, and G7Te. Likewise, the monogenean *Dactylogyrus* spp. was dominant in five of the 10 sampling points, with a dominance in the parasitic community higher than that reported by Blahoua Kassi et al. in 2019 [24] for monogeneans in the teleost cichlid *Tylochromis jentinki* (between *d* = 0.21 and *d* = 0.44), which shows low population balance of parasitic loads. The parasitic diversity indices can be indicators of the structure of the host fish population [25,26]. Likewise, it has been reported that the variations of the parasitic community that affect the fish can be attributed to the variations in the prevalence and abundance of dominant taxa [27].

The genus *Trichodina* spp. and the monogenean *Dactylogyrus* spp. in the fish farms they were in present 100% prevalence and 90% prevalence, respectively, while *Scyphydia* spp. and *Cichlidogyrus* spp. only prevailed in 10% of the production units.

### 3.3. Ecological Indices of Abundance and Intensity

The abundance values presented by the monogeneans in this study contrast with what was previously reported in tilapia farms in Yucatan, Mexico [28], where the abundance of the monogenean *Cichlidogyrus sclerosus* (73.84) was higher than those presented here (0.09–2.21). The ecological index of intensity is a parameter that allows for the characterization of the degree of infection (No. of parasites/No. of infected fish) in fish [29]. *Apiosoma* spp. and *Trichodina* spp. showed the highest intensity of infection. The intensity presented by *Trichodina* spp. was similar to that reported in tilapias cage-cultured in Brazil (96.4 ± 33.8) [20]. Data found for *Apiosoma* spp. suggest a high-intensity infection, since it was only identified in three sampling points, while the recurrent parasitosis by *Trichodina* spp. (present in the 10 sampling points) suggest a certain tolerance towards this pathogen. It has been reported that Nile tilapia can show resistance to other parasites such as *Gyrodactylus cichlidarum* in recurrent infections, and this is attributed to the response of the adaptive immune system in fish [30].

### 3.4. Physicochemical Water Quality

Point G1Ix showed the lowest value for pH with 6.46 ± 0.05. This value is close to that reported in concrete ponds for the farm of tilapia hatchlings in Costa Rica (6.60) [31]. The levels of CO_2_ may suggest that tilapia populations are at high densities, since at higher density, there is greater accumulation of this compound, which can influence the development of organisms [32]. Likewise, the analysis of the 10 water samples showed that the level of nitrogenous compounds is above that recommended for tilapia farming [33]. Yusni and Rambe in 2019 [34] reported ammonium levels of up to 1.13 and a positive relationship with the presence of monogeneans. This situation suggests a constant monitoring and control of nitrogenous compounds. Finally, the fish from *Valle del Mezquital* are farmed in so-called hard waters. The hardness values were like those reported by Cavalcante in 2014 (values of 58 to 529 mg L^−1^) [35], while the alkalinity was higher compared to the same study (53–104 mg L^−1^). However, the water from the farms of *Valle del Mezquital* maintained a ratio close to 1/1 between these parameters, an aspect that favors the growth and cultivation of these organisms [35].

### 3.5. Influence of Physicochemical Water Quality on Parasitoses

To determine the influence of physicochemical water quality on parasitosis, the sum of the products of the differences was obtained, divided by the sum of the products of the squared differences, by applying Equation (1), where x and y represent both data series compared [36].
(1)$$\frac{\sum_{i=1}^{n}\left(x_{i}-\tilde{x}\right)\left(y_{i}-\tilde{y}\right)}{\sqrt{\sum_{i=1}^{n}{\left(x_{i}-\tilde{x}\right)}^{2}}\;\sqrt{\sum_{i=1}^{n}{\left(y_{i}-\tilde{y}\right)}^{2}}}\;$$

The correlation of *Chilodonella* spp. (3.00% 2.54–3.46) and *Apisoma* spp. (12.00% (9.90–14.10) with the concentrations of nitrites (0.00–1.00 mg L^−1^) and ammonium (0.50–2.50 mg L^−1^) present in the water was similar to that reported by Ashmawy et al. in 2018 [37] about the association between the presence of ciliated protozoa (*Trichodina* spp.) and the concentrations of nitrogenous compounds (between 0.03 and 1.24 mg L^−1^). On the other hand, the Pearson matrix showed a positive correlation between the concentrations of nitrites (0.00–1.00 mg L^−1^) nitrates (10.00–33.33 mg L^−1^), and ammonium (0.50–2.50 mg L^−1^) (0.27 to 0.49 Pearson index), denoting the relationship among these compounds in the nitrification process of the culture environment [38]. The positive correlation observed between parasitic loads and nitrogenous compounds in this study has a precedent in infections by *Amirthalingamia macracantha*, *Clinostomum* spp., *Contracaecum* spp., *Tylodelphys* spp., *Argulus* spp. and *Neascus* spp. in the offspring of *O. niliticus*. [22]. This correlation may be due to these compounds degrading water quality and predisposing fish to parasitic infections [22,34]. Although in this study the monogenean *Dactylogyrus* spp. does not present a correlation with nitrogenous compounds, other studies such as the one by Ojwala et al. in Kenya in 2018 [39] have reported up to seven parasitic genera of tilapia positively related to the presence of nitrogenous compounds, including the monogenean *Cichlidogyrus hallis*. Finally, the positive correlation between populations of monogeneans has been previously reported between *Gussevia tucunarense* and *Gussevia arilla* in cichlid fish (*Cichla monoculus*), with a Spearman’s correlation coefficient of up to 0.69. These data suggest that there is no competition between these species [40]. It has been reported that in the presence of elevated nitrogenous compounds in the water, the cells rich in mitochondria and chlorides increase in count, in addition to the fact that the gills present lamellar fusion; this leads to a decrease in the number of functional Cl^−^/HCO3^−^ and to a state of alkalosis in fish, facilitating the colonization and proliferation of parasites such as monogeneans, which usually attack the anatomical branchial region. [41]. Correlation values in this study allow for the initial characterization of the parasitic loads behavior according to the particular characteristics of water quality present in the study area.

There was a correlation between the concentrations of nitrites and ammonium (0.50–2.50 mg L^−1^) with the parasites of *Chilodonella* spp. and *Apisoma* spp. The concentrations of nitrogenous compounds (between 0.03 and 1.24 mg L^−1^) are related to the prevalence of ciliated protozoa (*Trichodina* spp.) The correlation of monogenic and nitrogenous compounds has a precedent in the relationship of parasitic infections. The quality of the farm’s water is a determining factor in the health of aquatic organisms. The importance of water lies in the verification of the main physicochemical parameters. This verification is one of the key activities for the care of a health event.

## 4. Materials and Methods

### 4.1. Study Area

A total of 100 juvenile farmed hybrid tilapias (*Oreochromis niloticus* x *Oreochromis aureus*) were obtained for parasitic analysis and 30 water samples were taken from 10 sampling points distributed in four municipalities at *Valle del Mezquital* in the State of Hidalgo, Mexico (20°19′47.262″ N 99°15′22.633″ E). Figure 6 shows the sampling points and Table 4 shows the distribution of individuals by sampling point. The *Valle del Mezquital* region has a semi-arid climate with vegetation composed of xeric scrubs. The region is an Otomí-speaking indigenous area [42], where aquaculture practices are developed predominantly with the exploitation of tilapia in semi-intensive production systems that are mainly distributed in rural areas.

### 4.2. Sampling and Parasitological Analysis

The design of this study was cross-sectional, although the sampling was carried out on two different occasions six months apart as it was associated with the tilapia production cycles. Fish farms produce and market their product every six months; between cycle and cycle, the producers clean the facilities. Parasitic biodiversity was analyzed in each sampling, and it was found that the observed biodiversity was similar between each sampling. Tilapia sampling was carried out by convenience, according to the fishing opportunity. Sampling was carried out by local fishermen, whom carried out the process according to their customs and practices. Live fish were transported in 60 × 90 cm plastic bags with water and oxygen injection at a saturation >5 mg L^−1^ [43]. The fish were kept at room temperature (22 °C) until processing in the laboratory of parasitology, Institute of Agricultural Sciences, Autonomous University of the State of Hidalgo. Euthanasia was carried out by cranial puncture, and each individual was weighed and measured to determine the Fulton index [44] and then searched for ecto- [45] and endoparasites [46]. The handling of organisms was carried out in accordance with approval from the Ethics and Research Committee of the Autonomous University of the State of Hidalgo (Document ID: Comiteei.icsa 4/2021, Hidalgo, Mexico). The viscera and the intestinal content were observed by bright field optical microscopy. Photographs were obtained with a camera attached to a Zeigen WF10× microscope (nopCommerceCopyright © 2022 Zeigen Microscopios, CDMX, Mexico) to be processed later using ImageJ 64-bit Java 1.8.0 free software (National Institutes of Health, Bethesda, MD, USA). The recovered protozoans and monogeneans were deposited in 95% ethanol and kept at 22 °C [47]. Taxonomic identification was performed using standard keys. The analysis of the host-parasite relationship included the calculation of prevalence (No. of fish infected × 100/No. of fish collected), abundance (No. of parasites/No. of fish collected) and intensity (No. of parasites/ No. of infected fish), as well as the Simpson biodiversity index, which yields values between 0–1, where values close to 1 indicate greater equality between the parasitic populations that make up the community (Equation (2)), and the biodiversity of the community is expressed by the value of 1–*λ*. In the same way, the Berger-Parker index was calculated, which indicates the degree of dominance for the species with the highest frequency in the parasitic community. Its calculation yields values between 0–1, where values close to 1 show a high dominance (Equation (3)) [48,49]
(2)$$\lambda =\frac{\sum ni\left(ni-1\right)}{N\left(N-1\right)}$$
(3)$$d=\frac{N_{max}}{N}$$

### 4.3. Determination of Physicochemical Water Quality Parameters

Three samples of 200 mL of pond water were collected at each tilapia sampling point. Samples were deposited in glass jars at room temperature (22 ± 2 °C). The water pH and the concentration of nitrites (mg L^−1^ NO_2_–), nitrates (mg L^−1^ NO_3_ –), ammonium (mg L^−1^ NH_4_^+^), CO_2_ (mg L^−1^ CO_2_), hardness (mg L^−1^ CaCO_3_), and alkalinity were obtained using a Hanna ^®^ kit (Hannapro, SA de CV © 2022, CDMX, Mexico) by colorimetric determination [50]. The codes of the products making up the kit were HI98107 (pH), HI3873 (nitrites), HI3874 (nitrates), HI3824 (ammonium), HI3818 (CO_2_), HI3812 (Hardness) and HI3811 (Alkalinity) from Hanna ^®^.

### 4.4. Analysis of the Influence of Physicochemical Water Quality on Parasitic Biodiversity

The influence of physicochemical water quality on parasitic infections was analyzed by a Pearson’s correlation test using the free software RStudio (Boston, MA, USA). The index calculated by Pearson’s correlation yields a value between −1 to 1, where an index of 0 indicates that there is no relationship between the variables, 1 indicates a high similarity, values close to −1 indicate strong negative correlation, whereas the value of one variable increases, the other decreases. To calculate the index, two series of paired data are processed. 

## 5. Conclusions

The study of the ecological and biodiversity indices allows us to understand the interaction between parasite-host-environment components. In the present study, the levels of nitrogenous and water hardness showed a positive correlation with parasite infection; thus, management of organic matter and calcium carbonates can be areas of opportunity to improve tilapia production systems. Further studies are needed for the metagenomic and taxonomic identification of metazoan parasites to species level. This identification could not only be informative, but could allow elaborate distribution maps and, if necessary, identify new species.

## 6. Patents

There are no patents resulting from this work.

## Figures and Tables

**Figure 1 pathogens-11-01076-f001:**
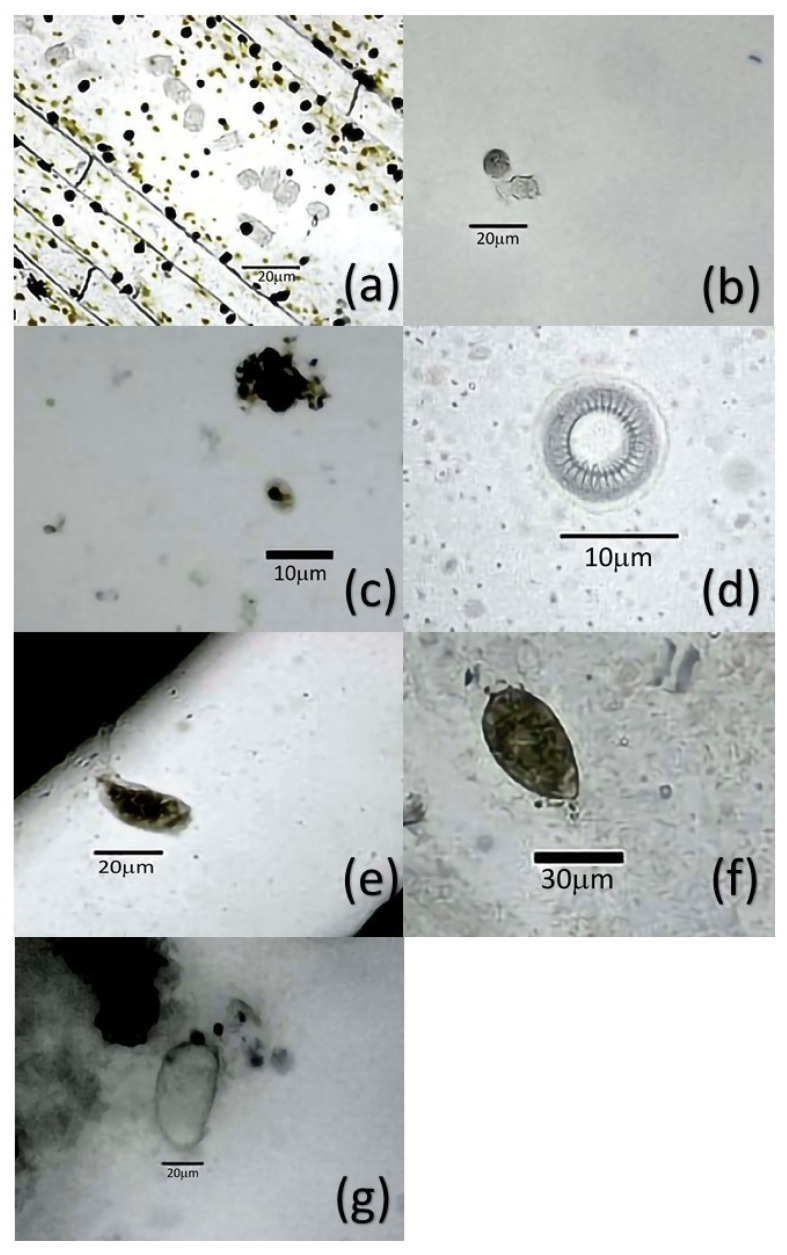
Protozoa in juvenile tilapia farmed at *Valle del Mezquital*. Panel (**a**) shows *Apiosoma* spp. recovered from gills and fins. Panel (**b**) shows *Chilodonella* spp. identified in gills and fins together with a specimen of *Apisoma* spp. Panel (**c**) shows *Costia* spp. identified in gill and fins. Panel (**d**) shows *Trichodina* spp. recovered from the gills. Panel (**e**) shows *Scyphydia* spp. identified in gills. Panel (**f**) shows *Tetrahymena* spp., specimens recovered from gills and Panel (**g**) shows *Oodinium* spp. recovered from gills.

**Figure 2 pathogens-11-01076-f002:**
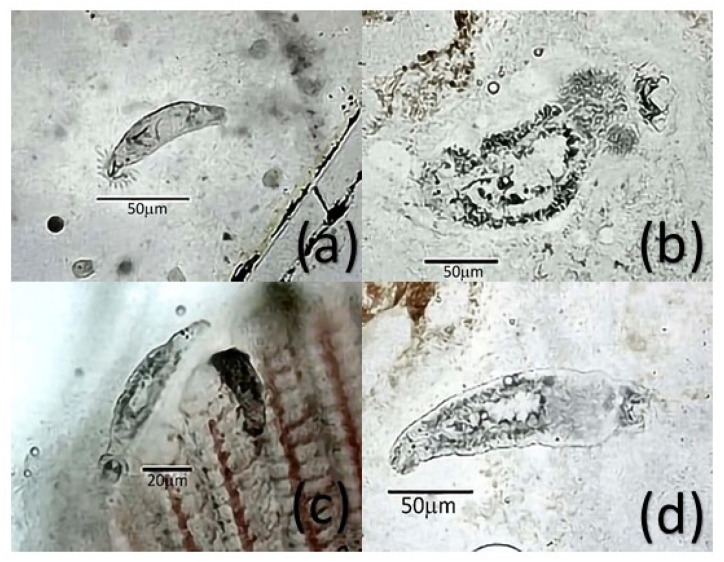
Monogeneans identified in juvenile tilapia farmed at *Valle del Mezquital*. Panel (**a**) shows *Cichlidogyrus* spp. Panel (**b**) shows *Dactylogyrus* spp. Panel (**c**) shows *Dawestrema* spp. Panel (**d**) shows *Gyrodactylus* spp. These monogeneans were recovered from the gill region.

**Figure 3 pathogens-11-01076-f003:**
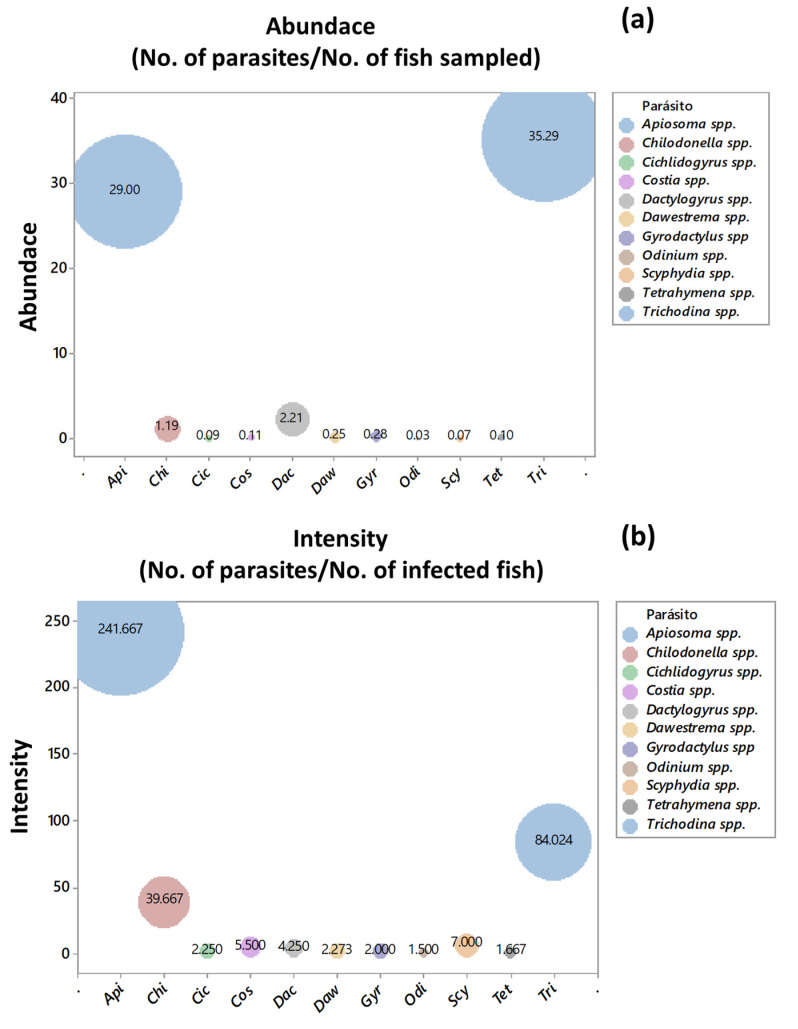
Globe plots for the ecological indices of abundance (**a**) and intensity (**b**). The indices were calculated for each parasitic genus recovered from farmed juvenile tilapia. Api: *Apiosoma* spp., Chi: *Chilodonella* spp., *Cichlidogyrus* spp., Cos: *Costia* spp., Dac: *Dactylogyrus* spp., Daw: *Dawestrema* spp., Gyr: *Gyrodactylus* spp., Odi: *Oodinium* spp., Scy: *Scyphydia* spp., Tet: *Tetrahymena* spp., Tri: *Trichodina* spp.

**Figure 4 pathogens-11-01076-f004:**
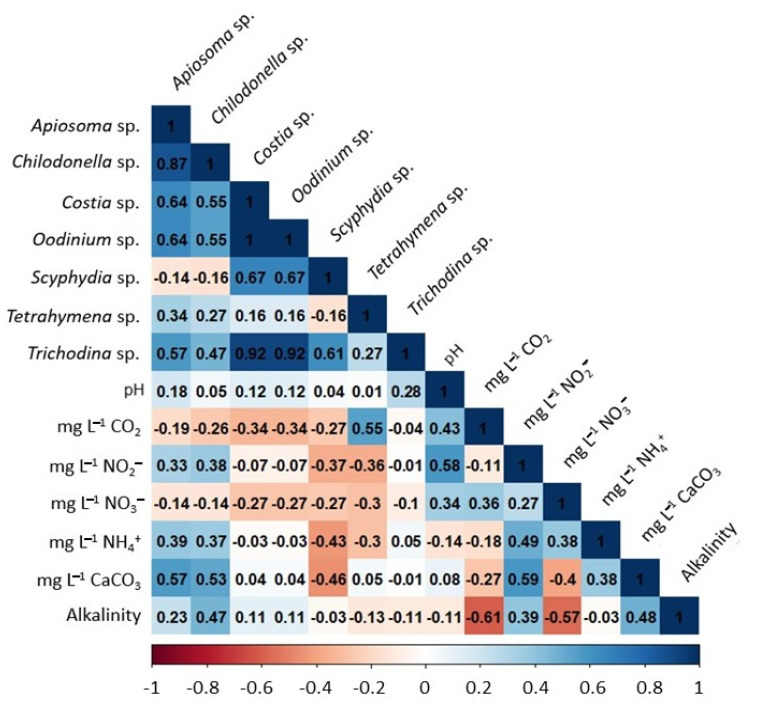
Correlation matrix between the prevalence of each protozoa genus determined in juvenile tilapias farmed and the physicochemical water quality characteristics at sampling points.

**Figure 5 pathogens-11-01076-f005:**
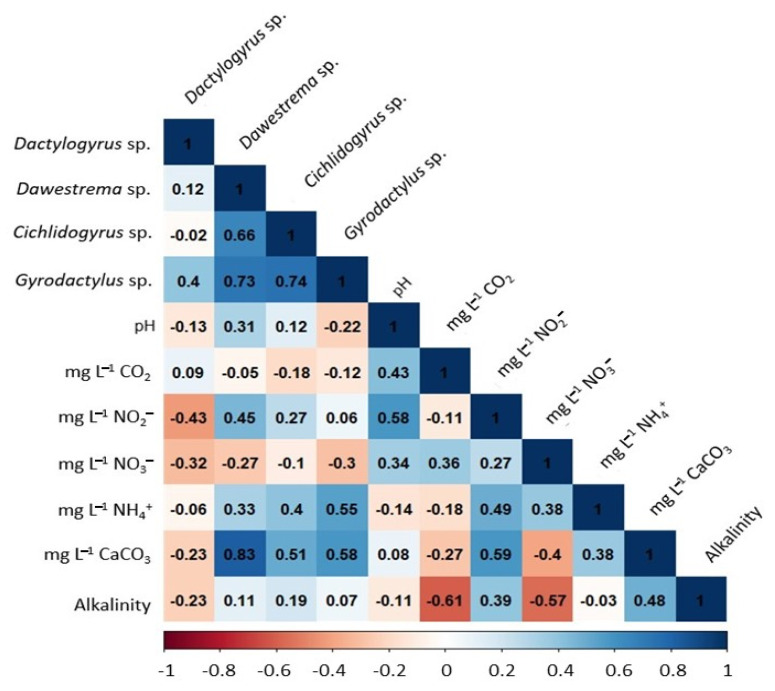
Correlation matrix between the prevalence of each genus of monogeneans determined in juvenile tilapias farmed and the physicochemical water quality characteristics at sampling points.

**Figure 6 pathogens-11-01076-f006:**
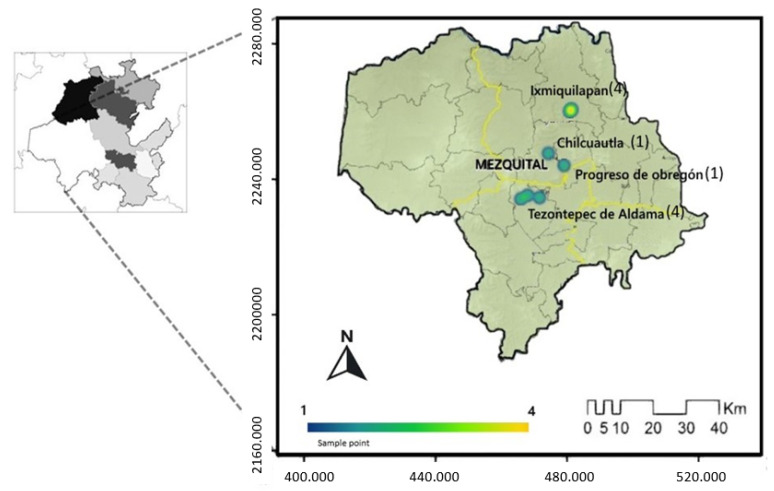
The *Valle del Mezquital* region is located in the State of Hidalgo, and can be seen in the enlarged image. The figure shows the four municipalities and, in parentheses, the number of samples per municipality.

**Table 1 pathogens-11-01076-t001:** Parasite prevalence of farmed juvenile tilapia in aquaculture production systems at *Valle del Mezquital*.

Parasite	Sampling Point										Parasite Prevalence ^1^ (IC 95%)
	G1Ix	G2 Ix	G3Ix	G4Ch	G5Ix	G6Te	G7Te	G8Te	G9Te	G10Pr	
**Protozoa**											
*Apiosoma* spp.	-	-	-	-	-	-	-	10	100	10	12% (9.90–14.10)
*Chilodonella* spp.	-	-	-	-	-	10	-	-	20	-	3% (2.54–3.46)
*Costia* spp.	-	-	-	10	-	-	-	-	10	-	2% (1.72–2.28)
*Oodinium* spp.	-	-	-	10	-	-	-	-	10	-	2% (1.72–2.28)
*Scyphydia* spp.	-	-	-	10	-	-	-	-	-	-	1% (0.79–1.21)
*Tetrahymena* spp.	-	-	40	-	-	-	-	-	20	-	6% (5.09–6.91)
*Trichodina* spp.	20	10	40	100	50	20	40	30	100	10	42% (39.76–44.24)
**Monogeneans**											
*Dactylogyrus* spp.	100	20	60	100	50	10	-	80	50	50	52% (49.65–54.35)
*Dawestrema* spp.	-	-	-	-	-	-	10	50	50	-	11% (9.60–12.40)
*Cichlidogyrus* spp.	-	-	-	-	-	-	-	-	40	-	4% (3.15–4.85)
*Gyrodactylus* spp.	40	-	10	-	-	-	-	30	60	-	14% (12.54–15.46)
**Total**											**86% (84.26–87.74)**

G1Ix: Ixmiquilapan farm 1. G2Ix: Ixmiquilapan farm 2. G3Ix: Ixmiquilapan farm 3. G4Ch: Chilcuautla farm 4. G5Ix: Ixmiquilapan farm 5. G6Te: Tezontepec de Aldama farm 6. G7Te: Tezontepec de Aldama farm 7. G8Te: Tezontepec de Aldama farm 8. G9Te: Tezontepec de Aldama farm 9. G10Te: Progreso de Obregón farm 10. ^1^ Prevalence per sampling point and calculated with 100 sampled fishes.

**Table 2 pathogens-11-01076-t002:** Parasite biodiversity by sampling point.

Sampling Point	Number of Parasitic Individuals	Richness of Parasitic Genera	Simpson Index	Berger–Parker Index
**G1Ix**	36	3	0.60	0.53 (*Dactylogyrus* spp.)
**G2Ix**	12	2	0.59	0.75 (*Dactylogyrus* spp.)
**G3Ix**	38	4	0.53	0.55 (*Dactylogyrus* spp.)
**G4Ch**	1954	5	0.07	0.95 (*Trichodina* spp.)
**G5Ix**	306	2	0.05	0.97 (*Trichodina* spp.)
**G6Te**	8	3	0.46	0.75 (*Trichodina* spp.)
**G7Te**	77	2	0.02	0.98 (*Trichodina* spp.)
**G8Te**	83	5	0.49	0.68 (*Dactylogyrus* spp.)
**G9Te**	4325	10	0.46	0.66 (*Apiosoma* spp.)
**G10Pr**	23	3	0.16	0.91 (*Dactylogyrus* spp.)
**Totals**	**6862**	**11**	**0.55**	**0.51 (*Trichodina* spp.)**

G1Ix: Ixmiquilapan farm 1. G2Ix: Ixmiquilapan farm 2. G3Ix: Ixmiquilapan farm 3. G4Ch: Chilcuautla farm 4. G5Ix: Ixmiquilapan farm 5. G6Te: Tezontepec de Aldama farm 6. G7Te: Tezontepec de Aldama farm 7. G8Te: Tezontepec de Aldama farm 8. G9Te: Tezontepec de Aldama farm 9. G10Te: Progreso de Obregón farm 10.

**Table 3 pathogens-11-01076-t003:** Physicochemical water quality parameters related to tilapia farming. The water samples analyzed correspond to the sampling points where tilapia samples were obtained for parasitological analysis ^1^.

Physicochemical Water Quality Parameters							
Sample	pH (*p* ≤ 0.001)	CO_2_ mg L^−1^ (*p* ≤ 0.001)	NO_2_^−^-N mg L^−1^ (*p* ≤ 0.001)	NO_3_^-^-N mg L^−1^ (*p* = 0.001)	NH_3_-N mg L^−1^ (*p* ≤ 0.001)	CaCO_3_ mg L^−1^ (*p* ≤ 0.001)	Total Alkalinity (*p* ≤ 0.001)
**G1Ix**	6.46 ± 0.05 ^e^	32 ± 1.73 ^c^	0.20 ± 3.4 × 10^−17 i^	13.33 ± 5.77 ^b^	2.50 ± 0 ^a^	307 ± 9.16 ^de^	361 ± 4.58 ^cd^
**G2Ix**	6.90 ± 0.10 ^d^	79 ± 4.58 ^c^	0.20 ± 3.4 × 10^−17 h^	20.00 ± 0 ^ab^	1.00 ± 0 ^cd^	273 ± 34.59 ^e^	317 ± 4.58 ^e^
**G3Ix**	7.03 ± 0.05 ^cd^	650 ± 17.32 ^a^	0.00 ± 0 ^j^	10.00 ± 0 ^b^	0.50 ± 0 ^d^	275 ± 18.33 ^e^	330 ± 3.00 ^de^
**G4Ch**	7.10 ± 0.10 ^bcd^	0 ± 0 ^c^	0.20 ± 3.4 × 10^−17 g^	10.00 ± 0 ^b^	0.66 ± 0.28 ^d^	191 ± 6.92 ^f^	370 ± 17.32 ^cd^
**G5Ix**	7.46 ± 0.05 ^a^	590 ± 121.24 ^a^	1.00 ± 0 ^e^	33.33 ± 11.54 ^a^	2.16 ± 0.28 ^ab^	197 ± 4.58 ^f^	255 ± 30.44 ^f^
**G6Te**	6.96 ± 0.05 ^cd^	46 ± 3.46 ^c^	1.00 ± 0 ^d^	13.33 ± 5.77 ^b^	1.50 ± 0.50 ^bc^	379 ± 15.39 ^c^	507 ± 27.49 ^a^
**G7Te**	7.03 ± 0.05 ^cd^	41 ± 3.46 ^c^	1.00 ± 0 ^c^	16.66 ± 5.77 ^b^	2.00 ± 0 ^ab^	428 ± 4.58 ^b^	378 ± 3.00 ^bc^
**G8Te**	7.30 ± 0.10 ^ab^	290 ± 17.32 ^b^	1.00 ± 0 ^b^	10.00 ± 0 ^b^	1.50 ± 0 ^bc^	517 ± 9.16 ^a^	369 ± 6.00 ^cd^
**G9Te**	7.16 ± 0.05 ^bc^	60 ± 3.00 ^c^	1.00 ± 0 ^a^	13.33 ± 5.77 ^b^	2.33 ± 0.28 ^a^	511 ± 7.54 ^a^	416 ± 4.58 ^b^
**G10Pr**	7.26 ± 0.05 ^ab^	60 ± 3.00 ^c^	1.00 ± 0 ^f^	13.33 ± 5.77 ^b^	1.16 ± 0.28 ^cd^	345 ± 3.00 ^cd^	469 ± 4.58 ^a^

Data show the mean of three replicates ± standard deviation. a–j indicate superscripts per column, different superscripts indicate statistically significant differences (*p* < 0.05) between samples. The sampling points refer to tilapia production aquaculture farms located in the State of Hidalgo.

**Table 4 pathogens-11-01076-t004:** Number and age of tilapias collected by municipality.

ID	Municipality	Latitude	Longitude	No. of Fishes	Fulton Index (Weight-Length Ratio)	Fish Stage
G1Ix	Ixmiquilpan	20.425037	−99.16348	10	1.52 ± 0.42	Offspring
G2Ix	Ixmiquilpan	20.423151	−99.165259	10	1.70 ± 0.16	Juvenile
G3Ix	Ixmiquilpan	20.424129	−99.164666	10	1.74 ± 0.33	Juvenile
G4Ch	Chilcuautla	20.305693	−99.228399	10	1.88 ± 0.28	Juvenile
G5Ix	Ixmiquilpan	20.418005	−99.170325	10	1.62 ± 0.18	Juvenile
G6Te	Tezontepec de Aldama	20.186341	−99.253102	10	1.86 ± 0.75	Juvenile
G7Te	Tezontepec de Aldama	20.192473	−99.285839	10	2.27 ± 1.30	Juvenile
G8Te	Tezontepec de Aldama	20.182211	−99.304982	10	2.82 ± 0.87	Juvenile
G9Te	Tezontepec de Aldama	20.18237	−99.304968	10	2.51 ± 1.00	Offspring
G10Pr	Progreso de Obregón	20.273165	−99.185246	10	2.46 ± 0.39	Juvenile
			**Totals**	**100**	**1.99 ± 0.44**	

## Data Availability

Not applicable.

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
