# Peer review of "Physicochemical Water Quality Influence on the Parasite Biodiversity in Juvenile Tilapia (Oreochromis spp.) Farmed at Valle Del Mezquital in the Central-Eastern Socioeconomic Region of Mexico"

_pathogens, 2022, doi:10.3390/pathogens11101076_

Round 1

Reviewer 1 Report

The manuscript entitled "Analysis of the influence of water quality on parasitic biodiversity in farmed tilapia (Oreochromis spp.) in the western region of the State of Hidalgo" is a well-documented manuscript that characterized the parasitic biodiversity in different Tilapia and tried to find a potential correlation with the water quality of each source. Overall they were able to show the potential influence on the dynamics of those infections. From the point of view of parasitology analysis, the authors did a gold standard work for characterization of the parasites in the population and did well in the statistical analysis.

Not impacting the acceptance of this paper here are my two cents: The only limitation is that the water quality was based just on its chemical characteristics of it. Environmental samples are well known to have also bacterial communities that can also affect this kind of analysis.

Metagenomic work would be beneficial for this study, not just to characterize the whole spectrum of the microbial community in those water samples but also to characterize potential low abundant parasitic infections potentially missed by the group.

Author Response

Good evening, the adjustments to the manuscript were made according to your observations, we sent the revised version, greetings

Reviewer 2 Report

Due to the implication (present and future) in the alterations in the parasitic biodiversity across fish farms depending on the quality of the water, this study is of especial relevance for mexican tilapia farming. The study is well deisgned and the manuscript well organized and writen. However, I would appreciate to answer to some concerns listed below: 

Line 30: is this mathematical correlation justified biologically?

Line 54: which is the aim of including crustaceans in the description?

Table 1, 2. Figure 3: due to the order of the manuscript where results are described previously to methodology, it would be appreciated to describe acronyms in the table legend to a better understanding of the table. At this regard, authors are encouraged to find a way of describing acronyms as G9Te or G4Ch at the beginning of the manuscript since when reading results are difficult to understand without going down to read the methods.

Point 2.6 and Figure 4: Authors need to include the significance in both the table and the text (when needed) since not including significance with a P<0.05 has no statistical value and it is not rigorous. Indeed, values included in the table, as well as in the text should show both the Pearson coefficient and significance difference given by the test to know whether de coefficient is significant or not. Moreover, asseverate that a correlation of 0.4 (eg.) is by one side a weak correlation and by the other not correct without knowing the significance probably driving to wrong or inaccurate conclusions.

Discussion: A separate point-by-point discussion as authors have described allows to focus on every index calculated but the discussion is only comparative.

However, there has not been a combined discussion of all the results that would allow to have a global vision of the present study. eg. What are the implications of the fact that the diversity, frequency of certain parasites is greater in certain areas than previous studies? Or the quality of the water altered when compared with previous studies? what could be the causes? Or the consequences?

Conclusions should be located just after the discussion for a better understanding of the discussion.

Line 483: “The study of their ecological and biodiversity indices allows us to understand the form of interaction between parasite-host-environment components.” Could the authors detailed a bit this form of interaction according the results obtained. 

Valle del Mezquital should be writen in italics

Author Response

(The authors gave the same response as above.)
